# Human Papillomavirus (HPV69/HPV73) Coinfection associated with Simultaneous Squamous Cell Carcinoma of the Anus and Presumed Lung Metastasis

**DOI:** 10.3390/v12030349

**Published:** 2020-03-22

**Authors:** Stephanie Shea, Marina Muñoz, Stephen C. Ward, Mary B. Beasley, Melissa R Gitman, Michael D Nowak, Jane Houldsworth, Emilia Mia Sordillo, Juan David Ramirez, Alberto E. Paniz Mondolfi

**Affiliations:** 1Department of Pathology, Molecular, and Cell-Based Medicine, Icahn School of Medicine at Mount Sinai, New York, NY 10029, USA; stephanie.shea@mountsinai.org (S.S.); stephen.ward@mountsinai.org (S.C.W.); mary.beasley@mountsinai.org (M.B.B.); Melissa.Gitman@mountsinai.org (M.R.G.); Michael.Nowak@mountsinai.org (M.D.N.); Jane.Houldsworth@mountsinai.org (J.H.); Emilia.Sordillo@mountsinai.org (E.M.S.); 2Grupo de Investigaciones Microbiológicas-UR (GIMUR), Departamento de Biología, Facultad de Ciencias Naturales, Universidad del Rosario, Bogotá D.C. 111221, Colombia; claudia.munoz@urosario.edu.co

**Keywords:** Human papillomaviruses, coinfection, HPV69, HPV73, carcinogenesis

## Abstract

Background: Human papillomaviruses (HPVs) have been linked to a variety of human cancers. As the landscape of HPV-related neoplasia continues to expand, uncommon and rare HPV genotypes have also started to emerge. Host-virus interplay is recognized as a key driver in HPV carcinogenesis, with host immune status, virus genetic variants and coinfection highly influencing the dynamics of malignant transformation. Immunosuppression and tissue tropism are also known to influence HPV pathogenesis. Methods: Herein, we present a case of a patient who, in the setting of HIV positivity, developed anal squamous cell carcinoma associated with HPV69 and later developed squamous cell carcinoma in the lungs, clinically presumed to be metastatic disease, associated with HPV73. Consensus PCR screening for HPV was performed by real-time PCR amplification of the L1 gene region, amplification of the E6 regions with High-Resolution Melting Curve Analysis followed by Sanger sequencing confirmation and phylogenetic analysis. Results: Sanger sequencing of the consensus PCR amplification product determined that the anal tissue sample was positive for HPV 69, and the lung tissue sample was positive for HPV 73. Conclusions: This case underscores the importance of recognizing the emerging role of these rare “possibly carcinogenic” HPV types in human carcinogenesis.

## 1. Introduction

Human papillomaviruses (HPVs) represent a subset of papillomaviruses that are known to infect human hosts and to contribute toward carcinogenesis as etiologic drivers in a wide variety of cancers [1]. Virtually all cervical cancers are associated with high-risk human papillomavirus infections [1,2]. In addition, the landscape of HPV-related neoplasia continues to expand, including a spectrum of head and neck, anal, vulvar, penile, and vaginal carcinomas [3,4].

Although most HPV infections clear soon after acquisition, infection persists in about 10% of cases [1]. The risk for persistence of HPV infection appears to be influenced by both biological and environmental factors, including age, ability to mount an immune response, hormonal status, dietary habits, tobacco and alcohol usage, and coinfection with other sexually transmissible agents [5]. The specific HPV type involved in infection is also critical for prognosis because HPVs are highly tissue-tropic, with some high-risk types linked to persistent infection and induction of carcinogenesis [2,3,4].

With over 100 types of HPV described to date, it is important to differentiate between low-risk and high-risk HPV types in order to appropriately guide medical management and care. The International HPV Reference Center at the Karolinska Institutet in Stockholm, Sweden, was first to categorize specific HPV types into “high-risk” (HR-HPV) and “low-risk” (LR-HPV) types according to their oncogenic potential [1,6,7]. The International Agency for Research on Cancer (IARC) has since further classified HPV types into risk groups: Group 1 defined as “carcinogenic to humans”; Group 2A as “probably carcinogenic to humans”; and Group 2B as “possibly carcinogenic to humans” [8]. The IARC Group 1 includes well-known oncogenic HPV types 16, 18, 31, 33, 35, 39, 45, 51, 52, 56, 58, and 59. Group 2A consists only of HPV 68. Group 2B includes HPV types 26, 53, 66, 67, 70, 73, and 82, based on human evidence, and HPV types 30, 34, 69, 85 and 97, based primarily on close phylogenetic analogy to other HPV types with evidence of carcinogenesis [9]. 

Herein, we present a case of a patient who, in the setting of HIV positivity, developed anal squamous cell carcinoma associated with HPV69 and later developed squamous cell carcinoma in the lungs, clinically presumed to be metastatic disease, associated with HPV73. This case underscores the importance of recognizing the emerging role of these “possibly carcinogenic” HPV types in human carcinogenesis. 

## 2. Material and Methods

### 2.1. Case Presentation

A 42-year-old man with history of HIV, on highly active antiretroviral therapy, and 7.5 pack-years of smoking, initially presented for perianal mass excision. The perianal lesion was verrucous, circumferential, and extensive throughout the anorectal perineum, causing high clinical suspicion for carcinoma. The mass was excised and pathology diagnosis confirmed that the specimen was positive for squamous cell carcinoma. HPV molecular testing was not performed at that time. 

Subsequently, two months after excision, the patient presented to the emergency room with severe perianal pain accompanied by serosanguinous to purulent discharge. Physical exam revealed ulceration along the prior excision site and a prominent right inguinal lymph node that had enlarged over the preceding weeks. In addition, computerized tomography (CT) of the pelvis suggested multiple areas of deep fluid collection and abscess formation. The patient was admitted to the hospital for intravenous antibiotic treatment and monitoring of infection in the setting of AIDS (absolute T cell count 239/mm3 with 8% T helper cells; viral load 116 copies per mL). He was also diagnosed with active hepatitis B during this admission (viral load >501,000,000 IU/ mL). Fine needle aspiration of the enlarged right inguinal lymph node demonstrated malignant cells, consistent with metastatic squamous cell carcinoma. The patient began treatment with chemotherapy (mitomycin/capecitabine) and radiation therapy and was discharged home with a plan to continue outpatient therapy.

Six months after the excision procedure, the patient returned to the emergency room with shortness of breath and was found to have new deep vein thrombosis on Doppler imaging. CT angiography of the chest ruled out pulmonary embolism but instead revealed eleven pulmonary nodules ranging from 3mm to 12mm in greatest dimension, scattered throughout both lungs and raising clinical concern for distant metastasis (Figure 1A,C,E). The patient eventually agreed to interventional radiology-guided biopsy. Multiple nodules were noted to have increased in size over a one-month time interval (Figure 1B,D,F), including the largest nodule, which had increased to 24mm from 6mm; this nodule was biopsied and diagnosed as squamous cell carcinoma. Over the next few weeks, the patient was noted to have new thrombi occluding the inferior vena cava, involving the right common iliac vein, and manifesting as a subsegmental pulmonary embolism. Ultimately, one month after lung biopsy, the patient passed away due to massive thrombosis within the heart. The family did not consent to autopsy.

### 2.2. Histopathology and Immunohistochemistry (IHC)

The anal excision and the lung biopsy specimens were fixed in 10% neutral-buffered formalin and embedded in paraffin. The paraffin-embedded tissues were cut into thin sections (4mm for anal lesion and 3mm for lung biopsy) and subjected to routine hematoxylin and eosin staining. Immunohistochemical staining was later performed using the primary antibody for p16 (E6H4, prediluted; Ventana Medical Systems, Inc, Tucson AZ, USA) on the Ventana Benchmark Ultra (Ventana Medical Systems, Inc, Tucson AZ, USA). Proper antigen retrieval was carried out according to the manufacturers’ guidelines. 

### 2.3. Polymerase Chain Reaction (PCR)

Genomic DNA was extracted from formalin-fixed, paraffin-embedded unstained tissue sections using the Maxwell^®^ 16 FFPE Plus LEV DNA Purification Kit on the Maxwell^®^ 16 MDx instrument (Promega, Madison, WI, USA) according to the manufacturer’s guidelines. 

Consensus PCR screening for HPV was performed by real-time PCR amplification of the L1 gene region, and the presence or absence of HPV16 and HPV18 genotypes was determined by amplification of the E6 regions with type-specific primers followed by High-Resolution Melting Curve Analysis, as described previously [10]. If HPV consensus PCR was positive but HPV 16-specific and HPV 18-specific PCR results were negative, Sanger sequencing of the L1 region was performed. The specific HPV genotypes present were determined using the Basic Local Alignment Search Tool (BLAST^®^), provided open source by the U.S. National Library of Medicine [11], to align the sequences obtained to the GenBank^®^ genetic sequence database. 

### 2.4. Data Analysis

After genetic sequence results were obtained through the Sanger method, an initial step to identify the viral types present in both samples was performed using a papillomavirus-specific BLAST® publicly available online at Papillomavirus Episteme (PaVE): the papillomavirus knowledge source (National Institute of Allergy and Infectious Diseases, Bethesda, MA, USA) [12]. 

Phylogenetic relationships amongst viral types within the Papillomaviridae family were evaluated using reference genome sequences for HPV, which were downloaded from within the PaVE online platform [12]. 

A multiple sequence alignment was performed via the Multiple Alignment using Fast Fourier Transform (MAFFT) program. Approximate maximum likelihood trees were inferred with FastTree 2.1 (https://bioweb.pasteur.fr/packages/pack@FastTree@2.1.10) under the Jukes-Cantor Substitution Model. The robustness of the nodes was evaluated using the bootstrap (BT) method with 1,000 replicates. Branches were rescaled and visualized using Interactive Tree Of Life V3 web tool (http://itol.embl.de). 

The phylogenetic analysis focused on reference genomes for Alphapapillomavirus because the two HPV types found within our patient were both members of this genus. In addition, two genomes each of the genera Betapapillomavirus and Gammapapillomavirus, both of which include HPV types that can infect humans, were included in phylogenetic analyses as outgroups. In the obtained phylogenetic tree, colors were assigned for the different species belonging to the genus Alphapapillomavirus and the risk classification was marked according to the parameters established by the IARC.

## 3. Results

### 3.1. Pathological Findings

The initial anal lesion had a condylomatous appearance both grossly and histologically. Microscopic examination revealed a papillary squamous proliferation with acanthosis, parakeratosis, and koilocytes. In addition, some sections demonstrated a transition from classic condyloma to squamous cell carcinoma (Figure 2A,B). Seven months later, biopsy of the largest, fast-growing, lung nodule revealed that pulmonary parenchyma was infiltrated by squamous cell carcinoma, keratinizing type (Figure 2C,D). 

### 3.2. Immunohistochemistry

Immunohistochemical staining for p16 exhibited strong and diffuse, > 70%, nuclear and cytoplasmic positivity in both specimens (Figure 3A,B). In anal squamous cell lesions, this pattern is highly suggestive of oncogenic HPV DNA integration into the host cells. Within the lung, p16 positivity by itself does not indicate HPV infection [13]; however, this finding may prompt additional testing for HPV in cases with suspicion for metastatic HPV-associated carcinoma. 

### 3.3. PCR and Phylogenetic Studies 

HPV screening performed by consensus PCR amplification of the HPV L1 gene region was positive in both anal and lung tissue samples, but HPV16 and HPV18 were not detected by type-specific PCR testing in either specimen. Sanger sequencing of the consensus PCR amplification product determined that the anal tissue sample was positive for HPV 69 (98% sequence identity), and the lung tissue sample was positive for HPV 73 (96% sequence identity). 

A phylogenetic tree was obtained for comparison of the HPV 69 and HPV 73 genetic sequences against reference HPV genomes, confirming genetic stratification within the alpha-5 and alpha-11 species groups, respectively (Figure 4). Both of these species groups fell within high-risk clades within the overall Phylogenetic map. 

## 4. Discussion

HPVs represent a subset of papillomaviruses, which are small nonenveloped icosahedral viruses that carry genetic material in double-stranded circular DNA [1]. Genetic sequencing and phylogenic comparisons of homologous genes have helped to elucidate appropriate subclassifications within the broad *Papillomaviridae* family, which encompasses viruses that infect a diverse range of host species [14]. HPV types that specifically infect humans are distributed among five taxonomic genera within the *Papillomaviridae* family: alphapapillomaviruses, betapapillomaviruses, gammapapillomaviruses, mupapillomaviruses, and nupapillomaviruses [1].

The squamous cell carcinomas of the anus and the lung from our patient were determined by Sanger sequencing to contain HPV69 and HPV73, respectively. Phylogenetic analysis confirmed that the respective genetic sequences fall within the alphapapillomaviridae species groups alpha-5 (HPV69) and alpha-11 (HPV73). Both the alpha-5 and alpha-11 species groups mapped within the oncogenic potential category, specifically into IARC Group 2B, defined as possibly carcinogenic to human [8,9]. 

At present, there is limited epidemiological data on the occurrence of HPV69 and HPV73 worldwide. This phenomenon may be explained by regional variability in the prevalence of HPV oncogenic types [15], as well as by variable performance of current detection and typing methods, which are generally designed to screen for the most common high-risk human-associated HPV genotypes [1]. 

In comparison to other high-risk HPV types, reports of HPV 69 and HPV 73 in squamous cell carcinomas have been rare. In a large retrospective cross-sectional study by de Sanjose et al. (2010), HPV69 and HPV73 each were found in <1% of invasive cervical cancer specimens [16]. Despite the rare rate of detection, there is already some evidence to suggest a carcinogenic role for HPV73. Munoz et al. (2003) compared HPV types present in women with squamous cell cervical carcinoma versus control women and found that HPV73 was associated with cancer, with an odds ratio >45 [17]. HPV73 has also been described in association with squamous cell carcinoma at other sites, including the esophagus [18] and the nail unit [19]. More recent research has demonstrated that HPV73 degrades p53 efficiently, providing biological evidence to support that HPV73 has immortalization capacity that may fuel carcinogenesis [20]. 

The detection of two different HPV strains at synchronous tumor sites represents a rare scenario. We acknowledge three possible explanations for this finding, listed here from least likely to most likely: (1) both carcinoma sites were primary tumors; (2) a second undetected primary tumor at another anatomic location led to metastastic lung lesions; and 3) the tumor arising in the anal condyloma was the primary tumor that later metastasized to the lungs, despite discrepant HPV types. Explanation 1 is unlikely, because clinically the lungs demonstrated multiple lesions, which is more compatible with presentation of metastatic disease. In addition, although HPV is occasionally identified within primary lung cancers, studies that have investigated this association have failed to establish a causal relationship in these instances [21,22]. Explanation 2 is also unlikely because the patient was being closely monitored with no evidence of additional lesions detected by physical examination or recent radioimaging studies, which included CT angiogram of the abdomen and pelvis in addition to the studies previously mentioned. Ultimately, we favor explanation 3 with the caveat that the original anal squamous cell carcinoma likely harbored multiple HPV types prior to metastasis and then demonstrated tissue tropism in the new metastatic site.

Coinfection with multiple HPV genotypes has been reported to occur with prevalence’s ranging from 4.4% to 73.8% worldwide [23,24]. Multiple HPV infection has been linked to a 31.8-fold higher risk for development of cervical cancer when compared to single HPV infection [23,25]. Moreover, intralesional HPV diversity has been proposed to be a determinant factor in HPV carcinogenesis [26,27].

HIV-infected individuals, in particular, demonstrate an increased likelihood for coinfection with multiple HPV types [5,28,29]. HIV infection appears to influence HPV pathogenesis while conferring an important risk factor to the development of HPV-associated cancers [5]. A recent study by Dube Mandishora et al. reported a significant association between HIV status and HPV clades for the most prevalent high-risk genotypes, suggesting that immunosuppression may influence persistence based on clade-specific features [26]. Furthermore, their study demonstrated that within an individual subject, HPV tropism and persistence at specific anatomical sites may be associated with particular intra-genotypic HPV variants [26]. 

For our patient, the finding of distinct HPV types at different tumor sites is consistent with reports of HPV diversity in HIV coinfected patients and variation in HPV types found in different anatomic sites in HIV-infected individuals. Pathophysiologically, differences between virus signatures observed in the anal and lung tumor sites from our patient may be explained on the basis of initial coinfection, followed by selection of a predominant virus type at each tumor site during the neoplastic process. 

In conclusion, although HPV69 and HPV73 have been classified by the IARC as group 2B, “possibly carcinogenic”, the case we report indicates that both these virus types can be associated with papillomavirus-induced carcinogenesis and metastasis. Our findings support a reconsideration of the current classification and inclusion of HPV69 and HPV73 among high-risk genotypes.

## Figures and Tables

**Figure 1 viruses-12-00349-f001:**
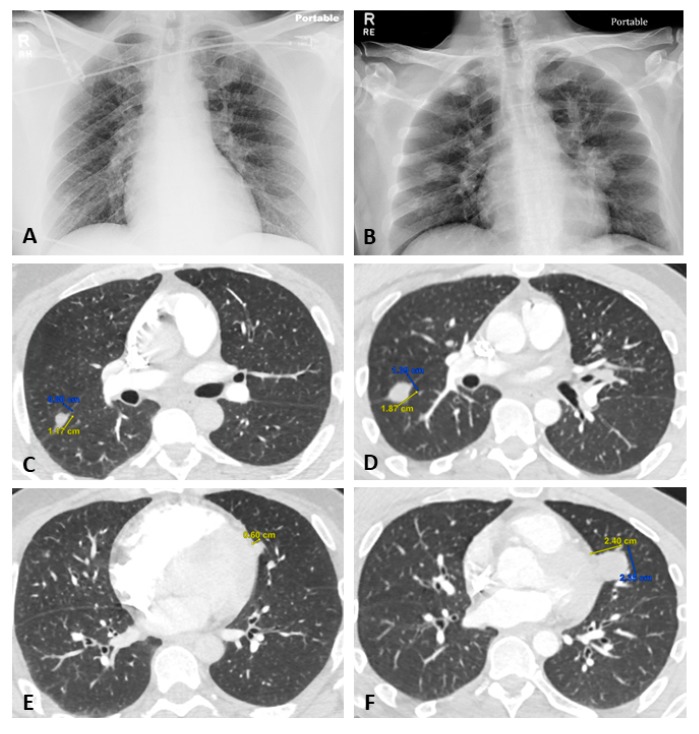
Imaging studies of the chest demonstrate increasing size of bilateral nodules between initial identification (**A**,**C**,**E**) and one month time interval (**B**,**D**,**F**). A-B, Chest x-ray. C-D, Right oblique fissure nodule visualized on computerized tomography (CT). E-F, Left upper lobe subpericardial solid nodule visualized on CT.

**Figure 2 viruses-12-00349-f002:**
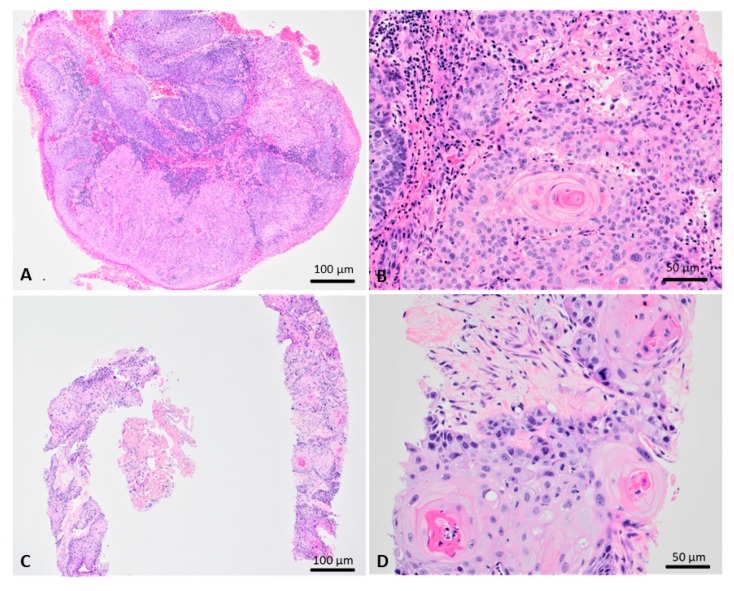
Microscopic findings of the anal lesion excision (**A**,**B**) and lung nodule biopsy (**C**,**D**). A, There is a notable distinction between the condylomatous area on the top/left and the carcinomatous area on the bottom/right (H&E; x40). B, The squamous cell carcinoma on the left demonstrates full-thickness dysplasia, keratinization, and invasive growth pattern (H&E; x200). C, The majority of the specimen is involved by squamous cell carcinoma (H&E; x40). D, The squamous cell carcinoma demonstrates marked cytologic atypia, keratinization, and invasive growth pattern (H&E; x200).

**Figure 3 viruses-12-00349-f003:**
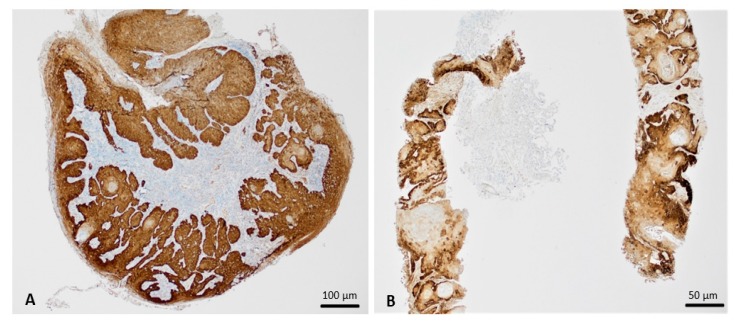
Immunohistochemical findings. (**A**), The anal lesion demonstrates strong, diffuse p16 positivity in both condylomatous and carcinomatous areas (x40). (**B**), The lung squamous cell carcinoma demonstrates strong, diffuse p16 positivity (x40).

**Figure 4 viruses-12-00349-f004:**
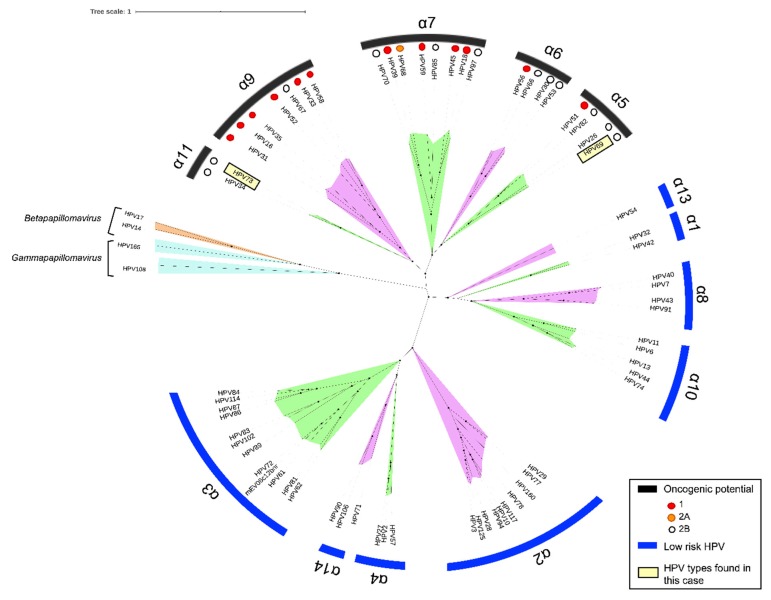
Alphapapillomavirus phylogenetic reconstruction based on whole genome sequences of reference Human papillomavirus (HPV) types available on PAVE database (https://pave.niaid.nih.gov/#explore/reference_genomes/human_genomes). The oncogenic potential have been marked according with the classification of The International Agency for Research on Cancer (IARC). HPV types found in our patient (HPV 69 and HPV 73 are boxed against reference HPV genomes).

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
