# Peer review of "Human Papillomavirus (HPV69/HPV73) Coinfection associated with Simultaneous Squamous Cell Carcinoma of the Anus and Presumed Lung Metastasis"

_viruses, 2020, doi:10.3390/v12030349_

Round 1

Reviewer 1 Report

This is an interesting case presentation that lends support to the oncogenic potential of HPVs 69 and 73, and I only had two comments.

  1. Did the authors attempt to re-evaluate the original anal biopsy sample to see if HPV 73 could be detected in support of their mixed infection model?
  2. The text only mentions PCR/sequencing on 1 lung lesion. Was only 1 lesion biopsied or was only 1 biopsy sample tested? If more than 1 lesion was biopsied it would have been interesting to see which HPV(s) were present.

Author Response

We thank the reviewer for his valuable comments and suggestions.

  1. Yes, we repeated the testing in the original anal sample when receiving results from the lung nodule. The results from the second test revealed HPV69 and failed to reveal the presence of HPV73. 
  2. Yes, only one pulmonary lesion was biopsied (the main nodule). We agree in that assessing multiple nodules would have been of greatest interest. Unfortunately, as mentioned in the manuscript, family declined autopsy and we were unable to perform post-mortem analysis of both condylomatous and pulmonary lesions.

Reviewer 2 Report

The case study is well written and will be of interest to others in the field.  Case report is comprehensively reported.

Author Response

We thank the reviewer for his interest in our work. Typos and spelling have been revised.